# Pre-operative systemic inflammatory response index influences long-term survival rate in off-pump surgical revascularization

**Tomasz Urbanowicz**[1]*, **Anna Olasińska-Wiśniewska**[1], **Michał Michalak**[2], **Bartłomiej Perek**[1], **Ahmed Al-Imam**[2,3], **Michał Rodzki**[1], **Anna Witkowska**[1], **Ewa Straburzyńska-Migaj**[4], **Michał Bociański**[1], **Marcin Misterski**[1], **Maciej Lesiak**[4], **Marek Jemielity**[1]

**1** Cardiac Surgery and Transplantology Department, Poznan University of Medical Sciences, Poznan, Poland, **2** Department of Computer Science and Statistics, Poznan University of Medical Sciences, Poznan, Poland, **3** Department of Anatomy and Cellular Biology, College of Medicine, University of Baghdad, Baghdad, Iraq, **4** 1st Cardiology Department, Poznan University of Medical Sciences, Poznan, Poland

* tomasz.urbanowicz@skpp.edu.pl

**Data Availability Statement:** Data are available from transplantologia@skpp.edu.pl for researchers

## Abstract

Coronary artery bypass revascularization is still the optimal treatment for complex coronary artery disease with good long-term results. The relation between inflammatory activation in the post-operative period and the long-term prognosis was already postulated. The possible predictive role of preoperative inflammatory indexes after the off-pump coronary artery bypass grafting technique on long term survival was the aim of the study. Study population included 171 patients with a median age of 64 years (59–64) operated on using off-pump technique between January and December 2014. Patients enrolled in the current study were followed-up for 8 years. We conducted a multivariable analysis of pre-operative and post-operative inflammatory markers based on analysis of the whole blood count. The overall survival rate was 80% for the total follow-up period, while 34 deaths were reported (30-days mortality rate of 1%). In the multivariable analysis, a pre-operative value of systemic inflammatory response index (SIRI) >1.27 (HR = 6.16, 95% CI 2.17–17.48, p = 0.012) revealed a prognostic value for long-term mortality assessment after off-pump surgery. Pre-operative inflammatory activation evaluated by systemic inflammatory reaction index (SIRI) possess a prognostic value for patients with complex coronary artery disease. The SIRI value above 1.27 indicates a worse late prognosis after off-pump coronary artery bypass (AUC = 0.682, p<0.001).

## Introduction

Diffuse atherosclerotic disease of the coronary arteries limits the daily activity and life span of patients due to insufficient blood supply of the myocardium. Concerning long-term results, coronary artery bypass revascularization is still the optimal treatment for complex coronary artery disease management [1]. The surgical revascularization allows for the complex provision

who meet the criteria for access to confidential data.

**Funding:** The authors received no specific funding for this work.

**Competing interests:** The authors have declared that no competing interests exist.

of coronary blood supply to the heart due to multiple grafts utility [2]. The arterial grafts provide superior results to venous ones which are commonly applied in clinical practice [3].

Coronary artery disease evolves from a chronic inflammatory process that leads to atherosclerotic changes influencing the survival rate [4–6]. The atherosclerotic plaques initiation and progression depend on an imbalance between pro-inflammatory and anti-inflammatory changes in the endothelium after triggering the inflammation reaction chain [7, 8].

The surgical revascularization can be performed with the utility of cardiopulmonary bypass (CPB) or with the off-pump technique [9, 10]. The omittance of CBP results in diminishing of the systemic inflammatory reaction which is the critical factor for long-term mortality risk [11]. The relation between inflammatory activation in the peri-operative period and the long-term prognosis was already postulated [8, 12].

Researchers proved the significant role of inflammatory indices in clinical practice in the prediction of the post-operative morbidity and mortality [13, 14]. The postulated indices include systemic inflammatory index (SII), systemic inflammatory response index (SIRI), and aggregate inflammatory systemic index (AISI). The clinical significance of post-operative indices in surgical coronary artery revascularization in mortality prediction has been already presented [15, 16]. Little is however known on the significance of the inflammatory indices on the long-term mortality after off-pump coronary bypass surgery.

We present the results of a single-center retrospective analysis of patients' survival after surgical revascularizations with the off-pump technique. The study aimed to evaluate pre-operative inflammatory markers for long-term outcomes prediction.

## Material and methods

The current study included 171 patients operated on in our Department between January and December 2014 as presented in Fig 1.

There were 71 (42%) patients with left main disease, 62 (36%) and 38 (22%) with diagnosis of three and two vessels disease, respectively. The indication for surgery was a chronic coronary syndrome. All patients were operated on by the experienced surgical team with the use the off-pump technique, and then followed up for a period of eight years. The exclusion criteria were concomitant procedures and haematological or oncological diseases. Moreover, patients with acute coronary syndromes were not taken into consideration.

Demographic and clinical data were collected. Although we focused in our study on inflammatory parameters obtained from whole blood count, the analysis was performed including additional laboratory parameters (troponin-I serum level, serum creatinine concentration, LDL cholesterol serum fraction) to obtain broader spectrum and distinguish significant factors. The history of previous peripheral vascular disease and events (PVE), including stroke and peripheral arterial disease (PAD) was analysed.

Pre-operative and post-operative (24 hours) whole blood results were obtained, including neutrophils, lymphocytes and platelets counts. Troponin-I levels were recorded after 4, 24 and 48 hours after surgery. The analysed inflammatory indexes were calculated as follows—systemic inflammatory index (SII)—quotient of neutrophils and platelets divided over lymphocyte counts; systemic inflammatory response index (SIRI)—quotient of neutrophil and monocyte divided by lymphocyte counts, and aggregate inflammatory systemic index (AISI)—quotient of neutrophils, monocytes and platelets divided by lymphocytes count.

C-reactive protein was not routinely measured as patients with infection suspicion were ruled out from the study. Postoperatively, C- reactive protein was not performed routinely except of patients presenting infection symptoms.

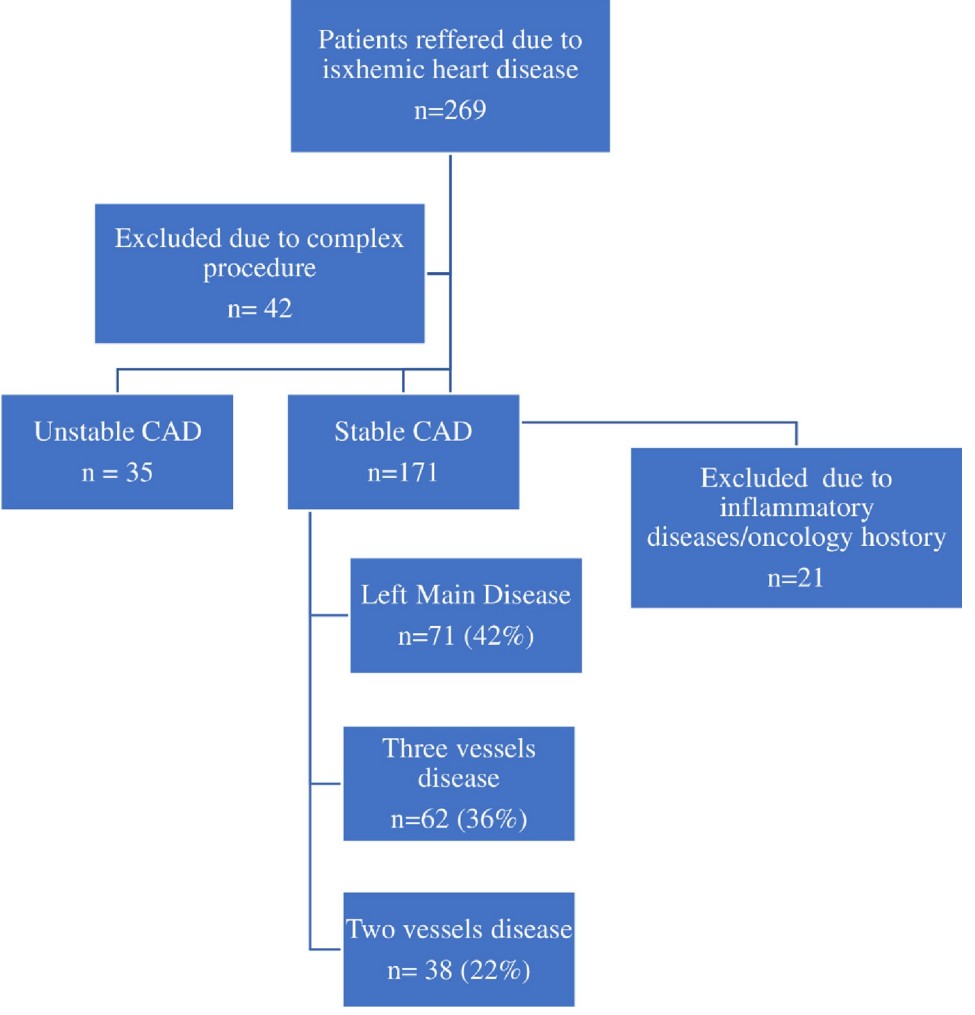

**Fig 1. Inclusion criteria for the study.**

Echocardiography was performed in all patients before and after the surgery, and at the discharge.

The surgical team operated on all patients under general anaesthesia with the off-pump technique through median sternotomy. The anastomoses were performed using intraluminal shunt application as the target coronary artery segment was stabilized by Octopus (Medtronic, USA). The anastomosis was performed with a single running 7–0 monofilament suture.

The mortality analysis was based on all-cause mortality, and the researchers confirmed the information concerning deaths in the National Healthcare Database.

The study was approved by the Institutional Ethics Committee (No 914/21 and date of approval: 24 November 2021) and respected the principles outlined in the Declaration of Helsinki.

## Statistical analysis

All continuous data were presented as medians and interquartile ranges Me ($Q_1$-$Q_3$) for non-normal distribution. A non-parametric test (Mann-Whitney) was used to compare the continuous variables. A receiver characteristic curve analysis was performed to reveal potential

predictors for mortality. A proportional hazard regression model was used to assess mortality risk factors. Both univariable and multivariable analyses (stepwise backward selection) were performed. The results were presented as hazard ratios (HR) and 95% confidence intervals (95% CI). The statistical analysis was performed using MedCalc statistical package; MedCalc® Statistical Software version 20.010 (MedCalc Software Ltd, Ostend, Belgium; https://www.medcalc.org; 2021). All tests were considered significant at $p < 0.05$.

## Results

The study group comprised 171 patients (152 (89%) males and 19 (11%) females) with a median age of 64 (59–64). The overall survival rate was 80% during the follow-up period, while 34 deaths were reported. The 30 days mortality rate was 1% (2 patients), and the mean hospitalization time was 11 +/- 2 days.

The study patients suffered from several co-morbidities, including arterial hypertension (152 (89%)), hypercholesterolemia (98 (57%)), diabetes mellitus (DM) (65 (38%)), peripheral arterial disease (PAD) (29 (17%)), history of stroke (19 (11%)) and chronic obstructive pulmonary disease (COPD) (16 (9%)). Preoperatively, the mean left ventricular diastolic diameter and left ventricular ejection fraction were 48mm (44–52) and 55% (50–60), respectively. There was no difference between the survivors and deaths group regarding the pre-operative clinical data except for hypercholesterolemia (p = 0.034) though the LDL cholesterol serum fraction was insignificant (p = 0.875), as presented in the Table 1.

The mean number of performed grafts was 2.2 +/- 0.1 and 2.2 +/- 0.2 in survivors and deaths groups, respectively.

Maximum post-operative Troponin-I were 1.97 (0.68–3.97) ng/ml in survivors vs 1.86 (0.46–4.92) in deaths subgroups (p = 0.667), respectively. We retrieved the complete blood count test results before surgery and 24 hours after (Table 2) and presented the significant pre-operative parameters in Fig 2.

The receiver-operating characteristic (ROC) curve analysis confirmed a significant effect of the pre-operative SIRI (AUC = 0.682, $p < 0.001$) with a sensitivity of 73.53% and specificity of 63.5%, and a cut-off value above 1.27, as presented in Fig 3.

### Univariable analysis

According to the univariate Cox regression analysis (Table 3), significant preoperative factors included PVE (summed stroke and peripheral artery disease) (HR = 2.70, 95% CI 1.17–6.26,

**Table 1. Characteristics of survivors and non-survivors' groups.**

| Pre-operative parameters | Survivors (n = 137) | Non-survivors (n = 34) | p-value |
|---|---|---|---|
| Clinical characteristics: | | | |
| 1. Age (years) median(Q1-Q3) | 63 (59–67) | 66 (63–73) | 0.020 |
| 2. Gender (M (%)/F (%)) | 121 (88%) / 16 (12%) | 31(91%) / 3 (9%) | 0.635 |
| Co-morbidities: | | | |
| 1. Arterial hypertension | 120 (88%) | 32 (94%) | 0.278 |
| 2. Hypercholesterolemia | 84 (61%) | 14 (41%) | 0.034* |
| 3. COPD | 10 (7%) | 6 (18%) | 0.637 |
| 4. History of stroke | 12 (9%) | 7 (21%) | 0.495 |
| 5. Peripheral artery disease | 20 (15%) | 9 (27%) | 0.099 |
| 6. Diabetes mellitus | 50 (37%) | 15 (44%) | 0.413 |

Abbreviations: AISI- aggregate inflammatory systemic index, F–female, M–male, n–number, SII–systemic inflammatory index, SIRI–systemic inflammatory response index, Q-quartile, WBC–white blood cell count. *—statistical significance.

**Table 2. Comparison of pre-operative and post-operative results between survivors and non-survivors groups.**

| Pre-operative parameters | Survivors (n = 137) | Non-survivors (n = 34) | p-value |
|---|---|---|---|
| Clinical characteristics: | | | |
| 1. Age (years) median(Q1-Q3) | 63 (59–67) | 66 (63–73) | 0.020 |
| 2. Gender M/F (%) | 121 (88%) / 16 (12%) | 31 (91%) / 3 (9%) | 0.635 |
| Preoperative laboratory results: | | | |
| 1. WBC x$10^9$/l (median(Q1-Q3)) | 7.47 (6.51–8.33) | 7.94 (7.55–9.88) | 0.058 |
| 2. Lymphocytes x$10^9$/l (median(Q1-Q3)) | 1.83 (1.47–2.27) | 1.83 (1.57–2.05) | 0.976 |
| 3. Neutrophils x$10^9$/l (median(Q1-Q3)) | 4.66 (4.02–5.73) | 5.45 (4.40–6.99) | 0.009* |
| 4. Hemoglobin mmol/l (median(Q1-Q3)) | 8.90 (8.20–9.45) | 8.55 (8.00–9.10) | 0.133 |
| 5. Platelets x$10^3$/l (median(Q1-Q3)) | 213 (188–259) | 220 (180–272) | 0.843 |
| 6. Monocytes x$10^9$/l (median(Q1-Q3)) | 0.42 (0.35–0.53) | 0.48 (0.40–0.63) | 0.021 |
| 7. SIRI (median(Q1-Q3)) | 1.11 (0.78–1.47) | 1.39 (1.18–2.01) | <0.001* |
| 8. SII (median(Q1-Q3)) | 562 (409–792) | 618 (498–1017) | 0.106 |
| 9. AISI (median(Q1-Q3)) | 235 (162–369) | 305 (213–500) | 0.031* |
| 10. Serum creatinine umol/L (median(Q1-Q3)) | 97 (81–121) | 99 (79–124) | 0.745 |
| 11. Troponin–I ng/mL (median(Q1-Q3)) | 0.02 (0.01–0.03) | 0.02 (0.01–0.03) | 0.901 |
| 12. LDL serum level mg/dL (median(Q1-Q3)) | 58 (51–66) | 60 (52–69) | 0.875 |
| Number of grafts | 2.2 +/- 0.1 | 2.2 +/- 0.2 | 0.897 |
| Postoperative laboratory results: | | | |
| 1. WBC x$10^9$/l (median(Q1-Q3)) | 8.51 (7.125–10.01) | 8.51 (7.13–10.01) | 0.955 |
| 2. Lymphocytes x$10^9$/l (median(Q1-Q3)) | 1.85 (1.53–2.34) | 1.77 (1.33–2.23) | 0.151 |
| 3. Neutrophils x$10^9$/l (median(Q1-Q3)) | 5.53 (4.03–6.55) | 5.53 (4.13–7.08) | 0.447 |
| 4. Hemoglobin mmol/l (median(Q1-Q3)) | 7.06 (6.6–7.55) | 6.75 (6.50–7.40) | 0.233 |
| 5. Platelets x$10^3$/l (median(Q1-Q3)) | 309 (244–355) | 255 (199–317) | 0.041* |
| 6. Monocytes x$10^9$/l (median(Q1-Q3)) | 1.02 (0.73–1.17) | 0.88 (0.64–1.18) | 0.446 |
| 7. SIRI (median(Q1-Q3)) | 2.46 (1.62–3.55) | 2.60 (1.73–5.51) | 0.329 |
| 8. SII (median(Q1-Q3)) | 764 (512–1109) | 864 (597–1059) | 0.633 |
| 9. AISI (median(Q1-Q3)) | 692 (424–1232) | 651 (484–1425) | 0.998 |

Abbreviations: AISI—aggregate inflammatory systemic index, F–female, M–male, n–number, SII–systemic inflammatory index, SIRI–systemic inflammatory response index, Q-quartile, WBC–white blood cell count. *—statistical significance.

p = 0.020), left ventricle diastolic diameter above 45mm (HR = 6.09, 95% CI 1.46–25.48, p = 0.013) and left ventricle ejection fraction below 50% (HR = 2.72, 95% CI 1.36–5.22, p = 0.005).

Significant preoperative laboratory markers for long-term prognosis prediction included—the neutrophils count above 5.2 (HR = 2.59, 95% CI 1.29–5.22, p = 0.007), monocytes count above 0.44 (HR = 2.83, 95% CI 6.13–11.39, p = 0.008), SIRI above 1.27 (HR = 1.39, 95% CI 1.72–7.69, p = 0.001) and AISI > 362 (HR = 2.28, 95% CI 1.13–4.76, p = 0.020). The postoperative echocardiographic results significant for mortality prediction included the left ventricle diastolic diameter above 48 mm (HR = 4.06, 95% CI 1.67–9.85, p = 0.002) and ejection fraction below 50% (HR = 2.56, 95% CI 1.28–5.13, p = 0.008).

## Multivariable analysis

We deployed multivariate analysis for parameters presenting as significant in univariable analysis, and novel data emerged concerning the significance concerning age (HR = 1.09, 95% CI 1.03–1.16, p = 0.005), COPD (HR = 5.24, 95% CI 1.84–14.91, p = 0.002), stroke (HR = 3.29, 95% CI 1.31–8.29, p = 0.012), and the preoperative left ventricle ejection fraction (HR = 3.28,

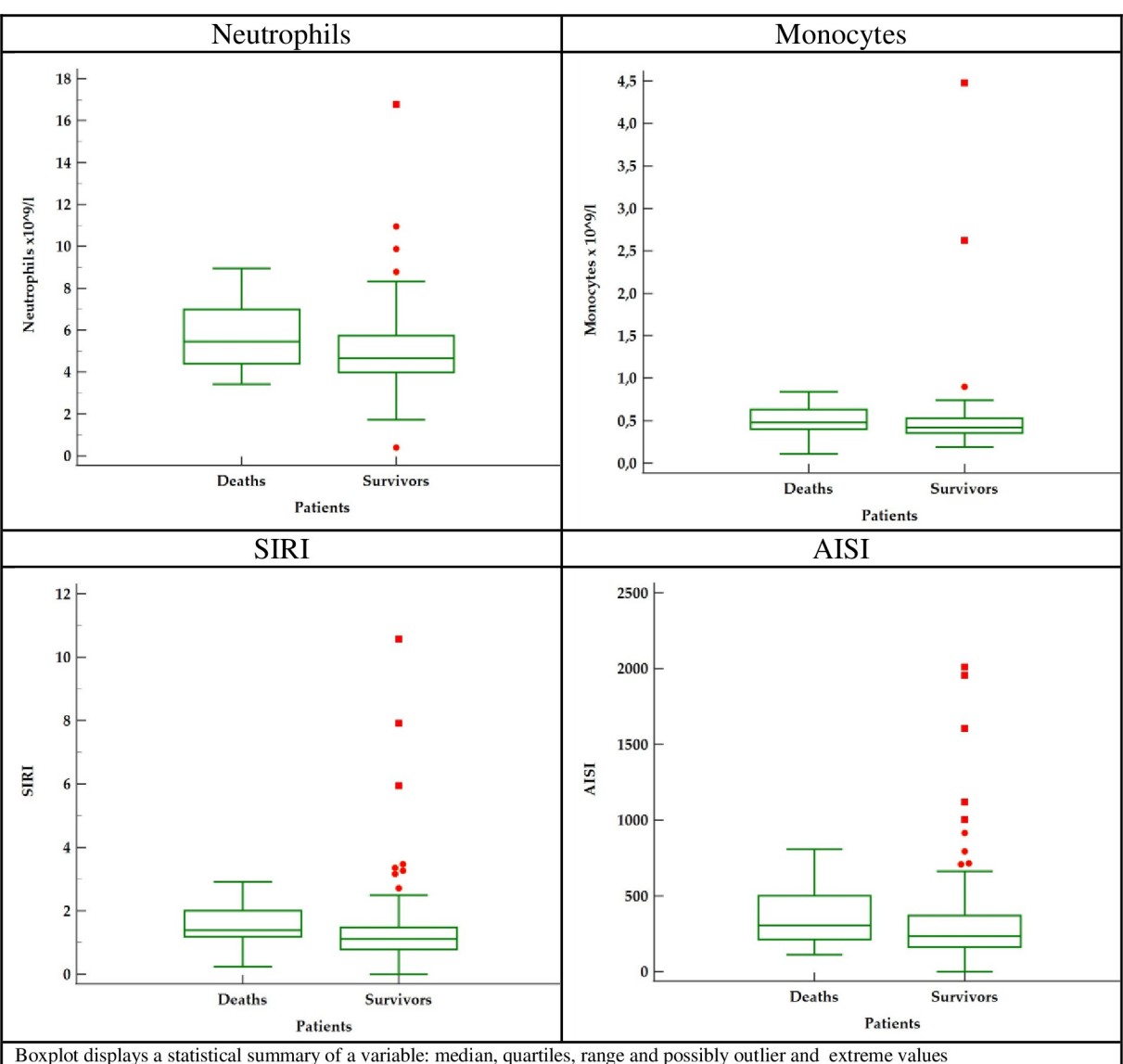

Boxplot displays a statistical summary of a variable: median, quartiles, range and possibly outlier and extreme values

**Fig 2. Box-plot 1.** Preoperative significant parameters.

95% CI 1.50–7.16, p = 0.003). Significant pre-operative laboratory markers obtained from whole blood count included SIRI above 1.27 (HR = 6.16, 95% CI 2.17–17.48, p = 0.001) as presented in Table 4.

## Discussion

The main finding of our study is the predictive value of pre-operative inflammatory activity characterized by systemic inflammatory reaction index (SIRI) for long-term mortality assessment. We present the results of the multivariable analysis of clinical and laboratory pre-operative factors in patients undergoing surgical revascularization due to complex coronary disease, suggesting that inflammatory reactivity possesses a high predictive significance for long-term results. Patients requiring surgical revascularization have an individual propensity for inflammatory activation that influences their post-operative survival [17, 18]. According to our

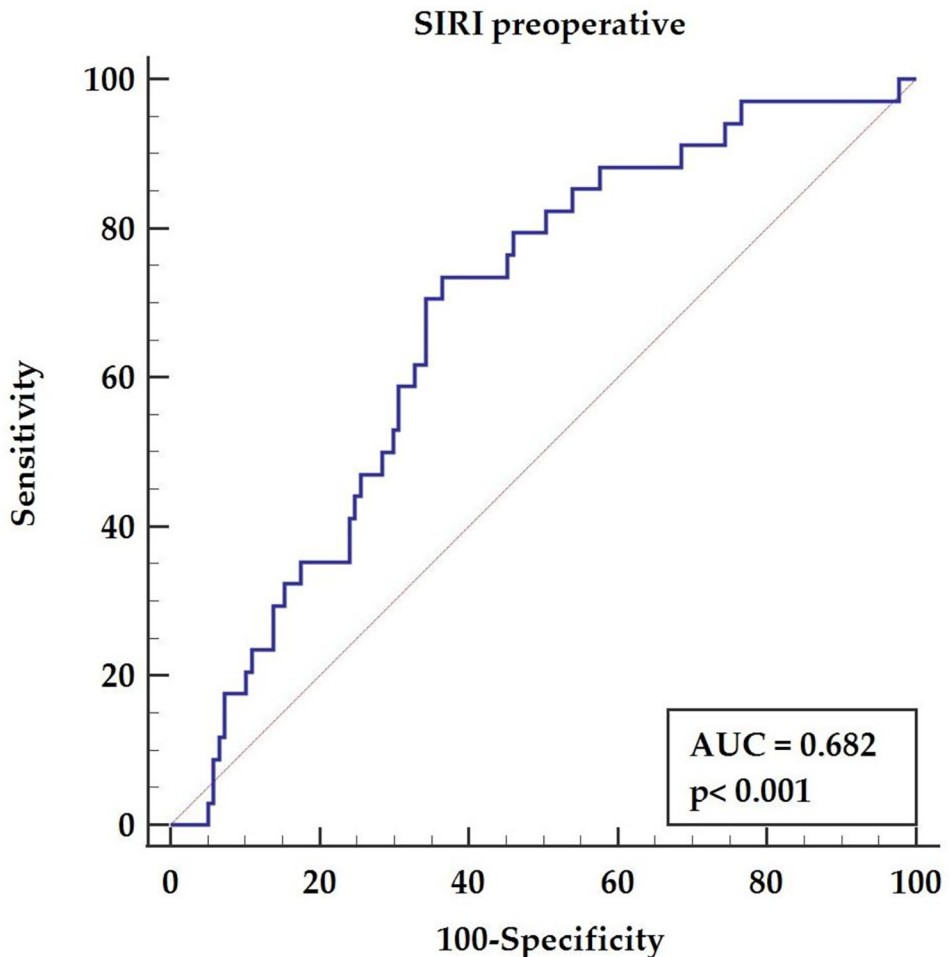

**Fig 3. The receiver-operating characteristic curve for the pre-operative value of systemic inflammatory response index.**

analysis those patients with more advanced inflammatory status before surgery, have higher risk of death. Therefore, we believe that this particular group of patients should undergo more scrutinous follow-up controls to improve long-term survival. Potentially, pharmacological treatment which may diminish the inflammatory activation in the atherosclerotic plaque, would be beneficial in this group of patients. Currently, statins have proved anti-inflammatory effect on the formation of atherosclerotic changes [19, 20].

Inflammatory processes activate atherosclerosis, or more accurately, its development and progression characterized by inflammatory markers and indices [21, 22]. Several previous studies [8, 23, 24] proved the relationship between inflammation, inflammatory cells activation and coronary artery disease occurrence. The degree of atherosclerotic changes in coronary arteries may be evaluated with the use of currently available diagnostic tools including coronary angiography, optical coherence tomography or cardiac magnetic resonance [25]. The measurement of inflammatory intensity influencing the atherosclerosis progression would be also beneficial. The inflammation reaction can be characterized by simple parameters obtained from the whole blood count [17, 18, 26, 27]. Those parameters, including the neutrophil to lymphocyte ratio (NLR) or monocyte to lymphocyte ratio (MLR), have predictive values for long-term mortality in off-pump patients after the surgical procedure [8]. The significance of

**Table 3. Univariable analysis results.**

| Parameters | HR | 95% CI | p-value |
|---|---|---|---|
| Stroke | 2.70 | 1.17–6.26 | 0.020* |
| Pre-operative echocardiographic results: | | | |
| 1. LVd | 1.07 | 1.02–1.12 | 0.009* |
| 2. LVd > 45 mm | 6.09 | 1.46–25.48 | 0.013* |
| 3. LVEF | 0.95 | 0.92–0.98 | 0.001* |
| 4. LVEF < 50% | 2.72 | 1.36–5.44 | 0.005* |
| Pre-operative laboratory results: | | | |
| 1. Neutrophils above 5.2 | 2.59 | 1.29–5.22 | 0.007* |
| 2. Monocytes > 0.44 | 2.83 | 2.31–6.13 | 0.008* |
| 3. SIRI > 1.27 | 1.39 | 1.72–7.69 | 0.001* |
| 4. AISI >362 | 2.28 | 1.13–4.76 | 0.020* |
| Postoperative echocardiographic results: | | | |
| 1. LVd > 48 mm | 4.06 | 1.67–9.85 | 0.002* |
| 2. LVEF < 50% | 2.56 | 1.28–5.13 | 0.008* |

Abbreviations: AISI—aggregate inflammatory systemic index, LVd–left ventricle diameter, LVEF–left ventricle ejection fraction, SIRI–systemic inflammatory response index.

our retrospective analysis is proposing SIRI as the pre-operative factor to predict worse late survival. We believe that searching for pre-operative factors which can reveal subgroups of patients prone to diminished long-term results is crucial. Our most valuable finding is the fact that easily available index (SIRI) may reflect higher mortality risk and therefore it is worth to include in daily practice in patients with coronary artery disease qualified for cardiac surgery.

The significance of inflammatory reactions as a possible trigger in broad spectrum of cardiovascular diseases is currently strongly underlined. The inflammatory activation presented by indexes in our study point out those patients who are more prone for complex artery disease development with secondary higher risk for worse outcomes. Previous studies revealed the relationship between inflammation and other cardiovascular risk factors and events.

**Table 4. Multivariable analysis results.**

| Parameters | HZ | 95% CI | p-value |
|---|---|---|---|
| Clinical: | | | |
| 1. Age | 1.09 | 1.03–1.16 | 0.005* |
| Co-morbidities: | | | |
| 1. COPD | 5.24 | 1.84–14.91 | 0.002* |
| 2. PVE | 3.29 | 1.31–8.29 | 0.012* |
| Preoperative echocardiography: | | | |
| 1. LVEF below 50% | 3.28 | 1.50–7.16 | 0.003* |
| Pre-operative laboratory parameters: | | | |
| 1. SIRI on admission > 1.27 | 6.16 | 2.17–17.48 | 0.012* |
| Postoperative laboratory parameters: | | | |
| 1. Platelets count | 0.99 | 0.98–0.99 | 0.002* |
| 2. Lymphocytes count | 0.35 | 0.13–0.92 | 0.034* |

Abbreviations: COPD- chronic pulmonary obstructive disease, PVE–peripheral vascular events in history, LVEF-left ventricle ejection fraction, SIRI–systemic inflammatory response index.

Szczepaniak et al. [28] presented the inflammatory link between hypertension and periodontitis. Inflammatory reaction related to nosocomial infection was described as a possible risk factor for worse presentation and outcomes after acute coronary syndrome [29]. Moreover, Fan et al. [30] identified a series of key genes closely related with inflammatory response and atrial fibrillation. The dietary intake influencing the inflammatory reactions in men that possess long-term results in increased risk for acute coronary syndromes was presented in Sut et al. [31] study. The relation between chronic inflammatory activation and a risk for left ventricular dysfunction was found in Kloch et al. [32] analysis.

The SIRI components as neutrophils, monocytes and lymphocytes play a significant role in atherosclerotic plaques formation and destabilization [33]. The initial destruction of endothelium cause monocytes adhesion and influx into intimal lawyer of the vascular wall with secondary secretion of cytokines and proteolytic enzymes [34]. The relation between hyperlipidaemia and neutrophilia [35] and monocytes activation [36] was presented in previous studies. The further modulation of monocytes/macrophages and foam cells at the site of lipids accumulation and regulated by neutrophils as these groups of cells interact in atherosclerotic plaques development [37].

The atherosclerotic plaque destabilization is of significant importance related to inflammatory cells activations. The role of innative inflammatory cells like monocytes [38] or indexes as possible markers as neutrophil to lymphocyte ratio [39] is postulated in plaques instability prediction.

Undoubtfully, both, on-pump and off-pump methods might cause an inflammatory response. The off-pump method of surgical procedures described in our analysis allowed us to minimize the influence of post-operative inflammatory response on our results. Several studies confirmed that the off-pump surgical revascularization technique omits the risk for inflammatory activation secondary to CBP administration [40–43]. In the Mirhafez et al. [41] study serum cytokines levels in off-pump coronary surgery were lower compared with on-pump method. Despite some disadvantages, the off-pump technique still possesses a profound protective value in high-risk patients [44, 45]. The main advantage of surgical revascularization is the time-related long-term survival rate [46]. Despite its invasive nature, the surgery gives superior results and places the surgical procedure as the optimal treatment for complex coronary artery disease [47–49] especially with ulitity of arterial grafts [50, 51].

Our analysis focused on pre-operative factors that may interfere with long-term prognosis. In multivariable analysis, the diminished left ventricle ejection fraction, combined with co-morbidities such as COPD and stroke, were presented significant which is consistent with the previous reports [52, 53]. Importantly, results from previous research focused on post-operative prognostic parameters [17]. To our best knowledge, our study is the first to reveal pre-operative inflammatory activation as patients' dependent characteristics that may interfere with post-operative results.

We noticed higher percentage of hypercholesterolemia in the survivor group. The serum LDL cholesterol fraction levels was insignificant, presenting effective therapy. However, this phenomenon was not statistically significant neither in the uni- nor in multivariable analysis. We shall point out that all patients referred for surgery reported long-lasting statin therapy, and the obtained results depended on the individual response to prescribed medication. Moreover, the results obtained for analysis were found irrelevant to hypercholesterolemia, as the target LDL serum levels were achieved suggesting more aggressive therapy in hypercholesterolemia group.

The long-term results following the off-pump coronary artery bypass grafting procedure depend on pre-operative, intraoperative, and post-operative factors [18]. Our analysis points out that among patients referred for off-pump coronary surgery, a subgroup is characterized

by excessive inflammatory activity that interferes with post-operative results and long-term survival.

The present study has several limitations. It is a retrospective, single-centre study limited to less than 200 cases. However, we included all consecutive patients who met the inclusion criteria, thus we showed the real-life cohort of patients. There is a gender discrepancy in presented population, but the consecutive patients were enrolled into the analysis and the same inequality is present in clinical practise. Moreover, we only analysed patients with stable complex coronary disease. We aimed to form the most homogenic study group, therefore, acute coronary syndrome was an exclusion criterium. Patients with concomitant diseases requiring surgical intervention were not included due to already proved worse results of combined procedures [54]. We believe the broader cohort of patients, preferably including multicentre research is necessary for further investigation.

## Conclusions

The systemic inflammatory response index is a prognostic factor for worse long-term outcomes after off-pump coronary artery bypass grafting in patients with chronic coronary syndrome. A pre-operative value of SIRI above 1.27 indicates patients with higher long-term mortality risk.

## Acknowledgments

Dr Ahmed Al-Imam is a participant of STER Internationalisation of Doctoral Schools Programme from NAWA Polish National Agency for Academic Exchange No. PPI/STE/2020/1/00014/DEC/02.

## Author Contributions

**Conceptualization:** Tomasz Urbanowicz, Anna Olasińska-Wiśniewska, Ewa Straburzyńska-Migaj.

**Data curation:** Anna Olasińska-Wiśniewska, Michał Rodzki, Marcin Misterski.

**Formal analysis:** Michał Michalak, Ahmed Al-Imam.

**Funding acquisition:** Marek Jemielity.

**Investigation:** Tomasz Urbanowicz, Anna Olasińska-Wiśniewska, Michał Michalak, Ahmed Al-Imam, Michał Rodzki, Anna Witkowska, Michał Bociański, Marcin Misterski.

**Methodology:** Tomasz Urbanowicz, Anna Olasińska-Wiśniewska, Michał Michalak, Bartłomiej Perek, Michał Rodzki, Anna Witkowska, Ewa Straburzyńska-Migaj, Marcin Misterski.

**Resources:** Michał Rodzki, Anna Witkowska, Michał Bociański.

**Software:** Michał Michalak, Ahmed Al-Imam.

**Supervision:** Marek Jemielity.

**Validation:** Michał Michalak, Bartłomiej Perek, Maciej Lesiak.

**Writing – original draft:** Tomasz Urbanowicz.

**Writing – review & editing:** Tomasz Urbanowicz, Anna Olasińska-Wiśniewska, Bartłomiej Perek, Ewa Straburzyńska-Migaj, Maciej Lesiak, Marek Jemielity.

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
