## [Decision Letter · Decision Letter 0]

19 May 2022

PONE-D-22-08151Pre-operative systemic inflammatory response index influences long-term survival rate in off-pump surgical revascularization.PLOS ONE

Dear Dr. Urbanowicz,

Thank you for submitting your manuscript to PLOS ONE. After careful consideration, we feel that it has merit but does not fully meet PLOS ONE’s publication criteria as it currently stands. Therefore, we invite you to submit a revised version of the manuscript that addresses the points raised during the review process.

ACADEMIC EDITOR:The manuscript underwent deep analysis and major revisions are required.

We look forward to receiving your revised manuscript.

Kind regards,

Antonino Salvatore Rubino, M.D., Ph.D.

Academic Editor

PLOS ONE

Journal Requirements:

Reviewers' comments:

Reviewer's Responses to Questions

**Comments to the Author**

1. Is the manuscript technically sound, and do the data support the conclusions?

Reviewer #1: Partly

2. Has the statistical analysis been performed appropriately and rigorously? 

Reviewer #1: Yes

3. Have the authors made all data underlying the findings in their manuscript fully available?

Reviewer #1: Yes

4. Is the manuscript presented in an intelligible fashion and written in standard English?

Reviewer #1: No

5. Review Comments to the Author

Reviewer #1: PONE-D-22-08151

Reviewer comment - The present manuscript aim to explore the possible predictive role of preoperative inflammatory indexes after the off-pump coronary artery bypass grafting technique on long-term survival. The study indicate that preoperative inflammatory activation, evaluated by systemic inflammatory reaction index (SIRI) possesses a prognostic value for complex coronary artery disease patients. A SIRI value above 1.27 indicates a worse late prognosis after off-pump coronary artery bypass.

The authors took up an interesting topic of research, but I have some comments to address them:

1. Materials and methods. A) The authors stated to include additional laboratory parameters other than blood count. Which ones?

Did they evaluated other inflammatory mediators such as C-reactive protein?

B) the definition of PVE and PAD is a bit confusing both in the test and in the tables. Sometimes the authors speak of PVE and others of stroke… Sometimes they use the abbreviations other the full name (Table 1. Point 4 and 5).

2. Results section: data is not well presented. I believe it is necessary to show a table with all the preoperative demographic and clinical characteristics of the study population.

The laboratory parameters, significantly different between the surviving and non-surviving groups, can be presented as box-plot images, instead of Table 2. Furthermore, the univariate table (Table 3) must contain all the variables that have been analyzed to better understand which enter the multivariable model.

I also would like to ask the authors to explain why they think hypercholesterolemia is higher in survivors than in non-survivors.

Did they also consider the different drug treatments?

6. PLOS authors have the option to publish the peer review history of their article (what does this mean?). If published, this will include your full peer review and any attached files.

Reviewer #1: No

---

## [Author Response · Author response to Decision Letter 0]

8 Jun 2022

Poznań, 7.6.2022

Dear Reviewer,

I would like to thank you for your valuable comments.

The answers are presented below.

If you have further suggestions, I would be more than happy to answer.

Kind regards

Tomasz Urbanowicz

Reviewer #1: PONE-D-22-08151

Reviewer comment - The present manuscript aim to explore the possible predictive role of preoperative inflammatory indexes after the off-pump coronary artery bypass grafting technique on long-term survival. The study indicate that preoperative inflammatory activation, evaluated by systemic inflammatory reaction index (SIRI) possesses a prognostic value for complex coronary artery disease patients. A SIRI value above 1.27 indicates a worse late prognosis after off-pump coronary artery bypass.

The authors took up an interesting topic of research, but I have some comments to address them:

1. Materials and methods. A) The authors stated to include additional laboratory parameters other than blood count. Which ones?

We included troponin level. The information is included in the Methods section

Did they evaluated other inflammatory mediators such as C-reactive protein?

Pre-procedural CRP was not routinely measured except from patients with signs of infection – those subjects were however excluded from the study group. We did not evaluated CRP levels postoperatively and did not correlated them with other markers as this marker was not routinely collected after surgery.

B) the definition of PVE and PAD is a bit confusing both in the test and in the tables. Sometimes the authors speak of PVE and others of stroke… Sometimes they use the abbreviations other the full name (Table 1. Point 4 and 5).

Thank you for this valuable comment – we unified the definitions. 

2. Results section: data is not well presented. I believe it is necessary to show a table with all the preoperative demographic and clinical characteristics of the study population.

Table 1 presents pre-operative characteristics of the study group 

The laboratory parameters, significantly different between the surviving and non-surviving groups, can be presented as box-plot images, instead of Table 2. 

Thank you for this valuable comments – we added box-plot images. 

Furthermore, the univariate table (Table 3) must contain all the variables that have been analyzed to better understand which enter the multivariable model.

Thank you for this valuable comments – we added all variables in the univariate table . 

I also would like to ask the authors to explain why they think hypercholesterolemia is higher in survivors than in non-survivors.

Did they also consider the different drug treatments?

We noticed higher percentage of hypercholesterolemia in the survivor group. However, this phenomenon was not statistically significant neither in the uni- nor in multivariable analysis. We shall point out that all patients referred for surgery reported long-lasting statin therapy, and the obtained results depended on the individual response to prescribed medication. We did not evaluated target serum levels of cholesterol’ fraction. Moreover, the results obtained for analysis were relevant for the time of surgery and the initial lipid profile characteristics was not available for the analysis. Therefore, the cholesterol level changes, and achievement of treatment goals were not evaluated and, in our opinion, should not be treated as a prognostic factor.

---

## [Decision Letter · Decision Letter 1]

29 Sep 2022

Pre-operative systemic inflammatory response index influences long-term survival rate in off-pump surgical revascularization.

PONE-D-22-08151R1

Dear Dr. Urbanowicz,

We’re pleased to inform you that your manuscript has been judged scientifically suitable for publication and will be formally accepted for publication once it meets all outstanding technical requirements.

Kind regards,

Vipin Zamvar

Academic Editor

PLOS ONE

Additional Editor Comments (optional):

Reviewers' comments:

Reviewer's Responses to Questions

**Comments to the Author**

1. If the authors have adequately addressed your comments raised in a previous round of review and you feel that this manuscript is now acceptable for publication, you may indicate that here to bypass the “Comments to the Author” section, enter your conflict of interest statement in the “Confidential to Editor” section, and submit your "Accept" recommendation.

Reviewer #1: All comments have been addressed

2. Is the manuscript technically sound, and do the data support the conclusions?

Reviewer #1: Yes

3. Has the statistical analysis been performed appropriately and rigorously? 

Reviewer #1: Yes

4. Have the authors made all data underlying the findings in their manuscript fully available?

Reviewer #1: Yes

5. Is the manuscript presented in an intelligible fashion and written in standard English?

Reviewer #1: No

6. Review Comments to the Author

Reviewer #1: The authors adequately answered to all points.

I wolud suggest them to check again the text.

There are still some word repetitions that prevent the flow and the readabilityof the manuscipt (e.g line 171 "concerning").

7. PLOS authors have the option to publish the peer review history of their article (what does this mean?). If published, this will include your full peer review and any attached files.

Reviewer #1: No
